# Impact of Urea Addition and Rhizobium Inoculation on Plant Resistance in Metal Contaminated Soil

**DOI:** 10.3390/ijerph16111955

**Published:** 2019-06-01

**Authors:** Guoting Shen, Wenliang Ju, Yuqing Liu, Xiaobin Guo, Wei Zhao, Linchuan Fang

**Affiliations:** 1State Key Laboratory of Soil Erosion and Dryland Farming on the Loess Plateau, Northwest A&F University, Yangling 712100, China; 18404984132@163.com (G.S.); 18970519821@163.com (Y.L.); aoei@nwafu.edu.cn (W.Z.); 2Institute of Soil and Water Conservation, Chinese Academy of Sciences, Ministry of Water Resources, Yangling 712100, China; juwenliang17@mails.ucas.ac.cn; 3University of Chinese Academy of Sciences, Beijing 100049, China; 4Agriculture Production and Research Division, Department of Fisheries and Land Resources, Government of Newfoundland and Labrador, Corner Brook, NL A2H 6J8, Canada; gxbguo@gmail.com

**Keywords:** heavy metal, legume-rhizobium, nitrogen, phytoextraction, plant resistance

## Abstract

Legume-rhizobium symbiosis has been heavily investigated for their potential to enhance plant metal resistance in contaminated soil. However, the extent to which plant resistance is associated with the nitrogen (N) supply in symbiont is still uncertain. This study investigates the effect of urea or/and rhizobium (*Sinorhizobium meliloti*) application on the growth of *Medicago sativa* and resistance in metals contaminated soil (mainly with Cu). The results show that Cu uptake in plant shoots increased by 41.7%, 69%, and 89.3% with urea treatment, rhizobium inoculation, and their combined treatment, respectively, compared to the control group level. In plant roots, the corresponding values were 1.9-, 1.7-, and 1.5-fold higher than the control group values, respectively. Statistical analysis identified that N content was the dominant variable contributing to Cu uptake in plants. Additionally, a negative correlation was observed between plant oxidative stress and N content, indicating that N plays a key role in plant resistance. Oxidative damage decreased after rhizobium inoculation as the activities of antioxidant enzymes (catalase and superoxide dismutase in roots and peroxidase in plant shoots) were stimulated, enhancing plant resistance and promoting plant growth. Our results suggest that individual rhizobium inoculation, without urea treatment, is the most recommended approach for effective phytoremediation of contaminated land.

## 1. Introduction

Heavy metal contamination in soil is considered a widespread problem due to growing anthropogenic disturbance and industrial development [1,2]. Copper (Cu) contamination is one of the most widespread and damaging types of heavy metal pollution in terrestrial ecosystems [3,4]. Although Cu is an essential microelement involved in plant biological processes, it is also highly toxic if absorbed in excess by plants. Excessive Cu in the soil can cause growth inhibition by interfering with a number of plant processes, including photosynthesis, respiration, oxygen superoxide scavenging, and cell wall metabolism and lignification [5]. Due to the impacts of Cu contamination on agriculture, it is crucial to remediate and restore Cu-contaminated soil.

Physical, chemical, and/or biological treatments are used to minimize Cu contamination [6,7]. The application of phytoremediation for contamination treatment has raised general concerns because it is both economical and environmentally friendly [6,8]. The efficiency of phytoremediation is dependent on two main variables: the concentration of metals in plants and plant biomass [9,10]. However, other factors, such as nitrogen (N) limitation in metal-contaminated soil, can also hinder the progress of plant processes and decrease the efficacy of phytoremediation [11,12,13]. To resolve this problem, N fertilizers are commonly applied to stimulate biomass accumulation by increasing N compounds (such as proline and glutathione) and chlorophyll content to promote photosynthesis. The application of urea has been found to increase the efficiency of metal phytoextraction in hyperaccumulators without altering metal concentrations: mainly by diluting metal toxicity through increased plant growth and shoot biomass [14,15]. There is also some evidence to suggest that N supply could enhance metal solubility and diffusion rates to the root surface, which stimulates metal accumulation in the shoots of hyperaccumulators [16,17]. N fertilization has also been demonstrated to enhance soil microbial structure and activity to improve phytoextraction efficiency in contaminated soil [18]. Further, N fertilization has been found to promote metal tolerance in plants by alleviating oxidative stress, decreasing photosynthesis inhibition, and increasing antioxidant enzyme activity through the production of free energy [19,20].

Legume-rhizobium symbiosis plays an important role in agriculture and ecological remediation. Legumes possess extensive root systems, fast growth rates, and the ability to obtain N through symbiotic relationships with rhizobium [21,22]. The symbiotic N fixation also could promote plant growth, thus increasing plant biomass under metal contaminated environment [11]. Additionally, rhizobia can improve both heavy metal resistance and phytoextraction capacity of symbiotic [23]. For example, extracellular polymeric substances (EPS) produced by rhizobium act as a first protective barrier to immobilize metal ions away from the cytoplasm, especially the loosely bound EPS, which have a rough surface and many honeycomb pores, promoting metal immobilization [24]. The ion-selective ATPase pumps have also been observed to conserve the metal transfer system in rhizobium [25,26,27]. Besides, rhizobia are beneficial to legumes through producing phytohormones, such as abscisic acid (ABA) and indole-3-acetic acid (IAA); the latter can alter the properties of root absorption, increase the amount of root exudates, and increase the number of plant growth regulator enzymes (1-aminocyclopropane-1-carboxylate (ACC) deaminase) [28,29]. Furthermore, rhizobium can stimulate the plant to produce antioxidant enzymes (including superoxide dismutase (SOD), peroxidase (POD), and catalase (CAT)) to scavenge the deleterious effects of heavy metals and re-establish homeostatic conditions [30,31,32,33]. In a previous study, without external N supply, N content increased with rhizobium inoculation accompanying by promoting related gene expression abundances of antioxidants enzymes [34]. Nevertheless, the metal resistance mechanism of legume-rhizobium symbiosis may be due to either/both N promoting efficacy or/and other internal traits that have not yet been elucidated.

In this study, we conducted a pot experiment to investigate the heavy metal resistance of the legume species *Medicago sativa* (*M. sativa*) in metals contaminated soil (mainly with Cu), with rhizobium inoculation (the metal-resistant rhizobium *Sinorhizobium meliloti* (*S. meliloti*)) and/or addition of urea. We chose urea as our N fertilizer as it serves as a primary N source, it can be effectively taken up by plants, and it is the most widely applied N fertilizer in agriculture on a global scale [35]. We hypothesized that N supply in rhizobium-inoculated plants would enhance the heavy metal resistance of plants in metal-polluted soil. Moreover, we speculated that the combination of rhizobium and urea would further improve plant resistance. The results of this study will increase our understanding of the metal resistant mechanisms of the legume-rhizobium symbiotic system in metal-contaminated soil and determine the applicability of phytoremediation to the restoration of contaminated land.

## 2. Materials and Methods

### 2.1. Pot Experiment

Surface soil (0–20 cm) was collected from the cultivated land nearby the abandoned Cu mine in the city of Huangshi, Hubei Province, China (30°07′ N, 114°91′ E). The soil samples were immediately transferred to the laboratory, and the selected physical and chemical properties of the soil were determined (Table 1). The samples were then air-dried and then passed through a 2 mm sieve. *M. sativa* seeds (provided by Beijing Rytway Ecotechnology Co., LTD, China) were surface-sterilized by 75% *v*/*v* ethanol for 3 min, followed by 10 min in 20% *v*/*v* NaClO (containing 8% available chlorine). The seeds were then washed with distilled water and germinated in a mixture of vermiculite and perlite (2:1) for 14 days. Approximately 20 pre-germinated seedlings were then transplanted into each pot (diameter 11 cm, and height 10 cm), and approximately 1000 g of the air-dried, sieved soil was added. To ensure optimal growth conditions for the plants, the moisture content was maintained at approximately 70% of the water-holding capacity by adding distilled water. Our pots were allocated to four treatments as follows: (1) *M. sativa*, control (M); (2) *M. sativa* and urea (MU); (3) *M. sativa* and *S. meliloti* (MS); (4) *M. sativa*, *S. meliloti,* and urea (MSU). Each treatment had three replicates.

A wild-type *S. meliloti* strain, a Cu-resistant bacterium, was inoculated in this study, which was isolated from the root nodules of *Medicago lupulina* in lead-zinc mine tailings in China [36]. The sample was deposited in the Agricultural Culture Collection of China (ACCC19736). *S. meliloti* was grown in a tryptone/yeast liquid medium (5 g tryptone, 3 g yeast extract, and 0.7 g CaCl_2_·2H_2_O L^−1^; pH 7.2) by shaking (150 rpm) at 28 °C. The growth of *S. meliloti* and the method of treatment was in accordance with the method of Kong et al. [28]. The rhizobium *S. meliloti* was able to endure higher concentrations of Cu in this study (Appendix A). After the plants had grown their first leaves, 20 mL of the bacterial cell suspension was sprayed into each pot three times, once a week, to inoculate the plants. Simultaneously, the equivalent amount of distilled water was added to the non-inoculated treatments. On the 60th day of plant growth, urea solution with a concentration of 5 mmol kg^−1^ (139.2 mg kg^−1^) was applied to the soil surface. After 90 days, the plants were harvested, and the soil samples were collected for further investigation.

### 2.2. Measurement of Soil Characteristics

The total content of metals in soils was determined by atomic absorption spectrophotometry (Hitachi Z2000, Hitachi, Japan) after the addition of 15 mL of the tri-acidic mixture (HCl, HNO_3_, HClO_4_) with a volume ratio of 1:3:1. The CaCl_2_-extractable Cu in soil was also determined by atomic absorption spectrophotometry, after shaking 5.0 g of soil with 25 mL of 0.1 M CaCl_2_ solution for 1 h [37]. Soil pH was measured using a pH meter (Model 225, Denver Instrument, Bohemia, NY, USA) with soil to water (non-CO_2_ deionized water) ratio of 1:2.5. The total N (TN) was measured using the Kjeldahl method [38]. Soil organic matter (SOM) was determined with a potassium dichromate external heating method [39]. The available phosphorus (AP) was measured using the 0.5 M NaHCO_3_ extraction-ammonium molybdate-antimony potassium tartrate and ascorbic acid spectrophotometric method (UV3200, Shimadu Corporation, Kyoto, Japan). Soil available potassium (AK) was extracted by 1.0 M CH_3_COONH_4_ solution and determined by ICP-AES (Perkin-Elmer Optima 3300DV, Perkin Elmer, Norwalk, USA).

### 2.3. Measurement of Shoot Height and Root Length, Plant Biomass, and Total Chlorophyll Content

The ruler was used to measure the shoot height and root length of *M. sativa* after they were harvested. The biomass of shoot and root was gauged with an analytical balance. The total chlorophyll content in leaves was determined by the method of Sobrino-Plata et al. [40]. Briefly, the leaves of *M. sativa* (0.20 g fresh weight) were powdered with liquid nitrogen, and then 25 mL 80% (*v*/*v*) aqueous acetone was added into the cuvette. The chlorophyll was extracted in the dark for 48 h until completely bleached and shook one time every 30 min. The total chlorophyll content was calculated from the absorbance of leaf extracts at 645 nm and 663 nm, respectively.

### 2.4. Measurement of Plant Cu and N Content

The harvested shoots and roots were washed with distilled water carefully. After samples were dried at 65 °C for 48 h, the plant samples were then separated into two portions. One portion was digested with H_2_SO_4_ and H_2_O_2_ for N concentration measured by flow analyzer. The other portion was prepared for determining the Cu content. Weighing about 0.30–0.50 g plant samples placed into digestion vessels and digested with an acid mixture (HNO_3_:HClO_4_ 4:1) were measured by an atomic absorption spectrophotometer (Hitachi Z2000, Hitachi, Japan).

### 2.5. Measurement of Plant Malondialdehyde (MDA), H_2_O_2_, and Oxygen Free Radical (OFR) Content

Lipid peroxide was evaluated by measuring the malondialdehyde (MDA) content [41]. The MDA content was measured using an MDA reagent kit (Suzhou Comin Biotechnology Co., Ltd. Suzhou, China). The absorbance of MDA was measured at 450, 532, and 600 nm, respectively. The production of oxygen free radical (OFR) was determined using an OFR reagent kit (Suzhou Comin Biotechnology Co., Ltd. Suzhou, China). The OD value of the solution applied was determined with a spectrophotometer (Mapada UV6300PC, Mapada, Shanghai, China) at 530 nm using NaNO_2_ as the standard curve. Hydrogen peroxide (H_2_O_2_) was measured using an H_2_O_2_ reagent kit (Suzhou Comin Biotechnology Co., Ltd. Suzhou, China). The content of H_2_O_2_ was detected by measuring the absorbance at 415 nm with a spectrophotometer. The above operations were performed all according to the manufacturer’s instructions.

### 2.6. Measurement of Plant Antioxidant Enzyme Activities

Fresh shoots and roots of plants were homogenized in an ice bath with 1 mL of extraction buffer (50 mM phosphate buffer solution, including 1 mM ascorbic acid and 1 mM EDTA) at 4 °C. The homogenate was centrifuged at 15,000 *g* for 15 min at 4 °C, and the supernatant was used for assaying antioxidant enzyme activities [37]. The activity of total superoxide dismutase (SOD, EC 1.15.1.1) was assayed by determining the ability to inhibit photochemical reduction of nitroblue tetrazolium (NBT) according to the instructions of a SOD reagent kit (Suzhou Comin Biotechnology Co., Ltd. Suzhou, China). One unit of SOD activity was defined as the amount of enzyme required to cause 50% inhibition of the reduction of NBT when determined at 560 nm. The activities of peroxidase (POD, EC 1.11.1.7) and catalase (CAT, EC 1.11.1.6) were assayed using their own specific reagent kit (Suzhou Comin Biotechnology Co., Ltd. Suzhou, China). POD activity and CAT activity were assayed by measuring the absorbance at 470 nm and 240 nm with a spectrophotometer, respectively.

### 2.7. Statistical Analysis

Two-way univariate analysis of variance (ANOVA) was conducted to analyze the effects of urea and rhizobium inoculation on properties of soil and plant with SPSS 20.0 (SPSS, Chicago, IL, USA). Duncan’s post-test (*p* < 0.05) was used for post hoc investigation. The Pearson correlation analysis was performed to measure the pairwise relationship between different variables. The heat maps of correlation between enzyme activities and the content of Cu and N, Cu uptake, and oxidative stress in the plant were performed using HemI software (Heat map Illustration, Version 1.0). All bar graphs were drawn using Origin 2018.

## 3. Results

### 3.1. Soil Physicochemical Properties

The physical and chemical properties of the untreated soil are presented in Table 1. Our soil samples were slightly acidic, and the Cu content of the collected soil far exceeded the national standard maximum recommendation for China (GB15618-1995), which is 400 mg kg^−1^. Following our experiment, the content of N and SOM increased after alfalfa planting. Both urea and inoculation had a significant effect on soil TN content (Table 2); the soil TN content significantly increased by 3.30%, 3.30%, and 7.44% in MU, MS, and MSU relative to M, respectively (*p* < 0.05). In terms of SOM, urea or inoculation significantly influenced the SOM, and these two factors had an interactive effect (Table 2). Specifically, the SOM was highest in the inoculated soil; SOM was 6.10% and 1.63% higher in treatments of MS and MSU relative to M, respectively (Table 2). The effects of urea or rhizobium and their interaction on the available Cu in soil were significant, and were enhanced by 4.64% and 15.8% in MS and MSU compared to M, respectively, but decreased by 22.5% after urea addition alone (Table 2). Additionally, only the urea treatment affected the total Cu content. Inoculation of rhizobium or/and urea addition induced a decrease in the total Cu content by 6.50%, 5.88%, and 6.04% in MU, MS, and MSU relative to M, respectively.

### 3.2. Plant Biomass, Chlorophyll, and N Content

Table 3 shows the plant phenotype, biomass, and chlorophyll content data of the different treatments. The addition of urea or rhizobium significantly promoted plant growth and increased biomass both in shoots and roots relative to the control. The urea and rhizobium inoculation had a significant interaction influence on root length, root biomass, and chlorophyll content. The largest values of root length, root biomass, and chlorophyll content were observed in MS, which were 90.3%, 51.6%, and 33.7% higher than the control, respectively. Furthermore, the co-application of rhizobium and urea had the highest plant height and shoot biomass among the different treatments. In MSU, the root length was shorter relative to the individual inoculation treatment. MSU also showed no significant difference in chlorophyll content relative to the control group. The factors of urea and rhizobium inoculation had an interactive effect on plant N content (Table 4 and Table 5). In shoots, the inoculation only had a significant effect on N content. In details, the shoot N content had increased by 2.20% and 7.11% in MS and MSU relative to M, respectively (Figure 1). In roots, the N content was 10.6% higher in rhizobium-inoculated plants relative to the control group. Similarly, the N content had also increased by 7.14% and 3.57% in MU and MSU relative to M, respectively.

### 3.3. Cu Content and Uptake in Plant Tissues

As shown in Table 6, we investigated the effects of urea application or/and rhizobium inoculation on plant Cu content. Both urea and inoculation influenced the Cu content in plants, and these two factors had an interactive effect. In comparison to M, the shoot Cu content significantly increased by 33.6%, 39.7%, and 45.8% in MU, MS, and MSU, respectively. The root Cu content was 38.0% higher and 8.13% lower in MU and MSU, respectively, relative to MS. Similarly, the Cu uptake in plants was also influenced by urea addition or rhizobium inoculation, but these two factors had no interactive effect on shoot Cu uptake. The root Cu uptake was 87.6%, 68.3%, and 51.1% higher in MU, MS, and MSU, respectively, relative to the control. Cu uptake in shoots was highest in MSU; shoot uptake increased by 41.7%, 69%, and 89.3% in MU, MS, and MSU, respectively, compared with the control. The correlation analysis showed that N content was positively correlated with Cu uptake in plants (*p* < 0.05), and only the shoot Cu uptake was positively correlated with N content in soil (*p* < 0.01) (Table 7). The calculated Cu bioconcentration factor was 1- and 1.1-fold higher in MS and MSU, respectively, relative to MU. Besides, rhizobium inoculation significantly influenced the translocation factor. The translocation factor was 1.4- and 1.6-fold higher in the treatments with rhizobium inoculation and urea-rhizobium combination plants, relative to urea application alone, respectively.

### 3.4. MDA, OFR and H_2_O_2_ Accumulation in Plant Tissues

In terms of MDA, both urea and inoculation had a significant effect on shoots, and two factors had an interactive effect (Table 4). The shoot MDA content decreased after the addition of both urea or/and rhizobium; a more pronounced decrease was observed with rhizobium inoculation (Figure 2A). In roots, urea had no significant effect on the MDA content, but there was a significantly interactive influence of two factors (Table 5). The value was significantly decreased by 36.6% and 16.7% in MS and MSU relative to the control group, respectively (*p* < 0.05). In shoots, the value was 16.3%, 30.4%, and 24.0% lower in MU, MS, and MSU than untreated plants, respectively. Interestingly, urea addition significantly stimulated the OFR content in roots, showing an increase of 4.90% and 9.74% in MU and MSU relative to M, respectively (Figure 2B). The accumulated OFR in the differently treated shoots was similar to the control group for neither urea nor rhizobium inoculation and had a significant effect (Table 4). H_2_O_2_ content (Figure 2C) significantly increased by 16.4%, 16.0%, and 26.9% in MU, MS, and MSU in roots compared with M, respectively. Additionally, in terms of shoot H_2_O_2_ content, the two factors urea and inoculation had an interactive effect on shoot H_2_O_2_ content (Table 4).

### 3.5. Plant Antioxidant Enzyme Activity

The inoculation had a significant influence on plant POD activity, and urea and rhizobium had an interactive effect (Table 4 and Table 5). Specifically, a significant reduction in root POD activity was observed after the application of rhizobium, showing a decrease of 34.5% and 13.5% in MS and MSU relative to M (Figure 3A, *p* < 0.05), respectively. The shoot POD activity increased by 22.4%, 45.9%, and 21.5% in MU, MS, and MSU relative to M, respectively. Both urea and rhizobium had a significant effect on plant CAT activity, and these two factors had an interactive effect. Post hoc Duncan comparison (*p* < 0.01) showed that CAT activity significantly increased in plant shoots after urea addition alone, but no obvious change was observed in either MS or MSU compared to control. In roots, rhizobium inoculation influenced the CAT activity, which increased by 14.7% and 34.2% in MS and MSU relative to M, respectively, and decreased by 61.7% in MU (Figure 3B). The root SOD activity was significantly influenced by the application of urea or rhizobium, and an interactive effect was also observed (Table 5), it was 19.6%, 71.6%, and 269% higher in MU, MS, and MSU relative to M, respectively (Figure 3C). In the urea-rhizobium combination treatment, SOD activity increased by 1.15-fold relative to individual inoculation. In shoots, urea interacted with rhizobium to alter SOD activity (Table 4). Post hoc Duncan comparison (*p* < 0.05) showed that only the combination of urea and rhizobium showed a significant increase in the SOD activity.

### 3.6. Correlation Analysis of Plant Oxidative Damage and Antioxidant Enzyme Activity

Figure 4 shows the correlations between oxidative damage, Cu content, Cu uptake, TN, and antioxidative enzyme activity in plants. In shoots, Cu content showed a strong negative correlation with MDA (*p* < 0.01) and a positive correlation with POD (*p* < 0.05). The MDA content showed a significant negative correlation with POD and Cu uptake (*p* < 0.01). Cu uptake was positively correlated with N content in shoots (*p* < 0.05). A negative correlation was observed between OFR and TN in shoots (*p* < 0.05). H_2_O_2_ was positively correlated with CAT (*p* < 0.05) and negatively correlated with SOD (*p* < 0.01). In roots, Cu content showed a strong negative correlation with CAT (*p* < 0.01). MDA was significantly positively correlated with a POD (*p* < 0.01), and both MDA and POD were strongly negatively correlated with N (*p* < 0.05). Cu uptake was positively correlated with N and H_2_O_2_ in roots (*p* < 0.05). There was a significant positive correlation between OFR and H_2_O_2_ (*p* < 0.01). OFR and H_2_O_2_ were also positively correlated with SOD (*p* < 0.01), and SOD and CAT were positively correlated (*p* < 0.05). Besides, we also observed that SOD in root was positively correlated with soil N content (*p* < 0.01) (Table 7).

## 4. Discussion

In our study, the result showed that the content of N and SOM increased after alfalfa planting in metals contaminated soil. This result confirmed the presence of root nodules and the assimilation of free N, thus increasing the N content in soil [42]. The reason for the increased SOM content could be that root exudates and other root-borne organic substances were released into the rhizosphere during the plant growth, as well as root hairs and fine roots were sloughed by root elongation [43]. Besides, it has been proved that the legume is loose growth and rots easily [44]. In terms of available Cu in soil, it was also increased after planting, suggesting that the existence of root was effective for the remediation of contaminated soil.

Heavy metal contamination is well known to reduce/inhibit plant growth [45,46]. Cu^2+^ ions, in particular, have a toxic effect on the primary reactions of photosynthesis through their impact on chlorophyll synthesis and electron transport [47]. Roots are the primary plant organs that are in direct contact with metal-contaminated soil and are thus generally more sensitive to metal toxicity [48,49]. Cu in the soil can cause an increase in ethylene biosynthesis in plants, which is known to cause a reduction in root length and induce apoptosis [25]. However, we observed a substantial increase in the length and biomass of both shoots and roots in inoculated plants. Root biomass in rhizobium-inoculated plants was also higher compared with those treated with urea (Table 3), and there was a negative interaction between urea and rhizobium. This result was in line with their effect on the N content in roots, suggesting that N plays an important role in promoting plant growth (Table 4). Additionally, a negative interaction was observed between urea and rhizobium in chlorophyll content, as evidenced by the fact that a combination of rhizobium inoculation and urea addition decreased plant chlorophyll content more than the only inoculation. The ANOVA analysis showed that urea is potentially harmful to inoculated-plant roots. In agreement, previous experimental results showed a decrease in the biomass of soybean nodules when grown with N, inferring that N addition reduces rhizobia performance [50]. These findings suggest that plants in symbiosis with rhizobium are more resistant to Cu, and the enzyme ACC deaminase may provide an explanation for these results [28]. ACC deaminase can help alleviate heavy metal toxicity and promote root growth by hydrolyzing ACC (the immediate precursor of the plant hormone ethylene) to NH_3_ and α-ketobutyrate, thereby reducing plant ethylene levels [51]. Interestingly, our results showed that root length was shortest when treated with urea alone, which was in accordance with the highest observed root Cu content (Table 4). The above observations suggest that adequate N supply can promote plant growth under metal pollution, but inhibits root growth under legume-rhizobium symbiosis.

In both shoots and roots of rhizobium-inoculated plants, the Cu content was significantly increased relative to the control. This could be attributed to the proteins produced from root nodules involved in antioxidant defense and metal detoxification, such as nicotinamine, phytochelatins, and metallothioneins, to detoxify metal [52]. In terms of the interaction of urea and rhizobium, there was an inverse influence on Cu content in shoots and roots. Substitutability, the Cu content was stimulated in shoots but decreased in roots with a combination of urea and rhizobium, which demonstrated the sensibility of the root in response to stress. Lower Cu uptake was observed in roots treated with both urea and rhizobium inoculation relative to inoculation alone (Table 4). There was a negative interaction, which was likely due to reduced root colonization from N fertilization [53]. The highest Cu uptake in the roots of urea treated plants (Table 2) could be explained by the fact that NH_4_^+^ could change the subcellular distribution (cell wall and vacuole could bind and sequester metals, thus limiting their translocation to shoot) [54] and chemical forms of metal (for example, metals integrated with pectates and protein generally have a lower migration) [54,55]. Additionally, our results showed that plant N was significantly positively correlated with Cu uptake in plants (Figure 4), and soil N was positively correlated with shoot Cu uptake, indicating that the addition of urea or/and rhizobium affected Cu uptake by altering the N content in plant tissues and soil. Furthermore, higher shoot translocation and bioconcentration factors were observed in inoculated plants compared with urea-treated plants (Table 4), suggesting that inoculation facilitated Cu phytoextraction [23]. Urea treatment had the lowest transfer coefficient value, similar to NH_4_^+^-N. It has been demonstrated that, when NH_4_^+^ is present as the sole N source, the internal transfer of Cd^2+^ from root to shoot in rice is inhibited through suppression of the metal transporter genes *OsNramp5* (the major transporter associated with Cd/Mn uptake in rice) and *OsHMA2* (transporter of Cd^2+^/Zn^2+^ from root to shoot) [56]. We conclude that rhizobium inoculation alone improved the efficiency of phytoextraction due to the observed increase in biomass accumulation and the small variability of Cu concentration.

Some studies have shown that the increase in MDA content is evidence for oxidative toxic stress in plant tissues [57,58]. Our results showed that the MDA content in plant significantly decreased in all inoculated plants relative to the control group, particularly for single inoculation treatments (Figure 2A). With urea addition in inoculated plants, MDA content increased both in shoots and roots (inhibitory effect). The correlation analysis showed that MDA content was negatively correlated with N content in plant roots, suggesting that high N content could lower the level of MDA and reduce plant stress. Additionally, the MDA content was strongly negatively correlated with Cu uptake in shoots (Figure 4A), probably indicating that rhizobium inoculation through alleviating oxidative stress promoted plant growth, thus increasing the Cu uptake. There was no interaction between urea and rhizobium inoculation of OFR (Table 5). The highest value of OFR in roots was observed with the application of urea and rhizobium, indicating that external N supply could induce stress to the inoculated plant roots. In plant shoots, the relationship between the content of N and OFR was also negatively correlated. These results indicate that N supply in legume-rhizobium symbiosis reduces oxidative stress under Cu pollution. This was also consistent with our first hypothesis that N supply in rhizobium-inoculated plants enhanced the resistance. These findings were in accordance with a previous study, which demonstrated that the addition of biodegradable chelant helped to detoxify metals and/or increase plant tolerance by elevating its external N concentration [18]. However, higher contents of root MDA and OFR in a combination of urea and rhizobium reflect that urea addition in symbiotic plants might depress the effect of oxidative stress scavenging.

Plants possess antioxidant enzyme systems to prevent cellular damage from superfluous ROS formed from environmental stress [34,59]. Cu content in plant shoots was highest when treated with both urea and inoculation, followed by inoculation alone, and lowest with urea treatment alone. Urea application enhanced enzyme activity, including CAT and SOD, suggesting that both enzymes are vital constituents in the defense mechanism of plant shoots (Table 4). There was an interactive effect of urea and rhizobium on Cu content in plant shoots, that is, the Cu content was slightly higher than with inoculation alone. However, POD activity had increased significantly when treated with inoculation alone relative to the combination treatment. This observation suggests that inoculation alone triggers the rise of POD activity in plant shoots. Besides, rhizobium had no effect on shoot SOD activity (Table 4). The above indicated that urea and rhizobium inoculation has a different resistance response to Cu stress in plant shoots. CAT activity in plant roots tended to decrease after the addition of urea; the inhibitory effects of urea on CAT activity could be ascribed to the observed increase in plant root Cu content. Our correlation analysis also revealed a negative correlation between CAT activity and Cu content in plant roots. Meanwhile, our results showed that both urea and rhizobium inoculation had a significant effect on the activities of SOD and CAT in roots (Table 5). Furthermore, compared with the control group, inoculation significantly enhanced the activities of SOD and CAT, indicating that both enzymes play a central role in improving plant root resistance. POD activity differed to CAT and SOD activities in roots whereby no significant change was observed with Cu content. A positively interactive effect of urea and rhizobium was observed on antioxidative enzyme activity in roots. Although the plants that were treated with a combination of urea and rhizobium showed highest SOD and CAT activities in roots (particularly for SOD activity), the highest content of OFR and H_2_O_2_ in roots, likely produced by external N and excessive Cu, was also observed. This observation contradicts our second hypothesis that urea addition would improve inoculated-plant resistance. Rhizobial N-fixation is energy intensive, and the process requires photosynthates from the legume. Thus, when N is sufficient, the plant becomes dominant in taking up N from the soil, which likely enhances SOD activity [60,61]. In our experiment, the soil N content was positively correlated with SOD activity in plant roots, which was supplied by urea or/and rhizobium. On the other hand, the high concentrations of both H_2_O_2_ and SOD activity in inoculated plant roots is intuitive, as SOD is central in converting superoxide radicals to both H_2_O_2_ and molecular oxygen [62]. Based on our results, we propose that rhizobium inoculation alone is the most effective method to promote plant growth, mitigate oxidative stress, and activate antioxidant enzyme activity. We propose that this method be used to enhance phytoremediation efficiency.

## 5. Conclusions

This study investigates the effects of both urea or/and rhizobium symbiosis on metals resistance of the legume species *M**. sativa* in contaminated soil mainly with Cu. Our results showed that the application of urea (as an N supply) or/and rhizobium increased the Cu uptake in shoots and roots by altering N contents in plants and soils. The application of urea or/and rhizobium promoted plant growth, as confirmed by high chlorophyll contents and plant biomass, by alleviating metals-induced oxidative damage and stimulating antioxidant enzyme activities. The negative interactive influence was observed between the addition of urea and rhizobium, especially in roots, which decreased the N content and increased the oxidative stress more than the rhizobium inoculation alone. Thus, we conclude that individual rhizobium inoculation was the most effective method to improve plant resistance in our experiment. N-fixation in legume-rhizobium symbiosis played a key role in enhancing Cu phytoextraction and plant resistance. Our results contribute to the understanding of the metal resistance mechanisms involved in the legume-rhizobium symbiotic system and suggest that individual rhizobium inoculation is the recommended approach for effective and efficient phytoremediation of contaminated land.

## Figures and Tables

**Figure 1 ijerph-16-01955-f001:**
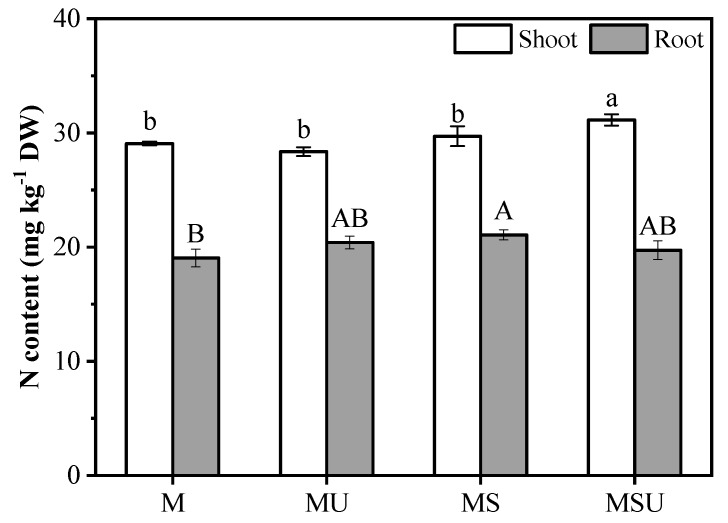
The effect of rhizobium on the total N contents in shoot and root. M: *M. sativa*, MU: *M. sativa* + urea, MS: *M. sativa* + *S. meliloti*, MSU: *M. sativa* + *S. meliloti* + urea. Bars are standard error (*n* = 3). Different letters stand for significant difference (*p* < 0.05) with Duncan’s post-test.

**Figure 2 ijerph-16-01955-f002:**
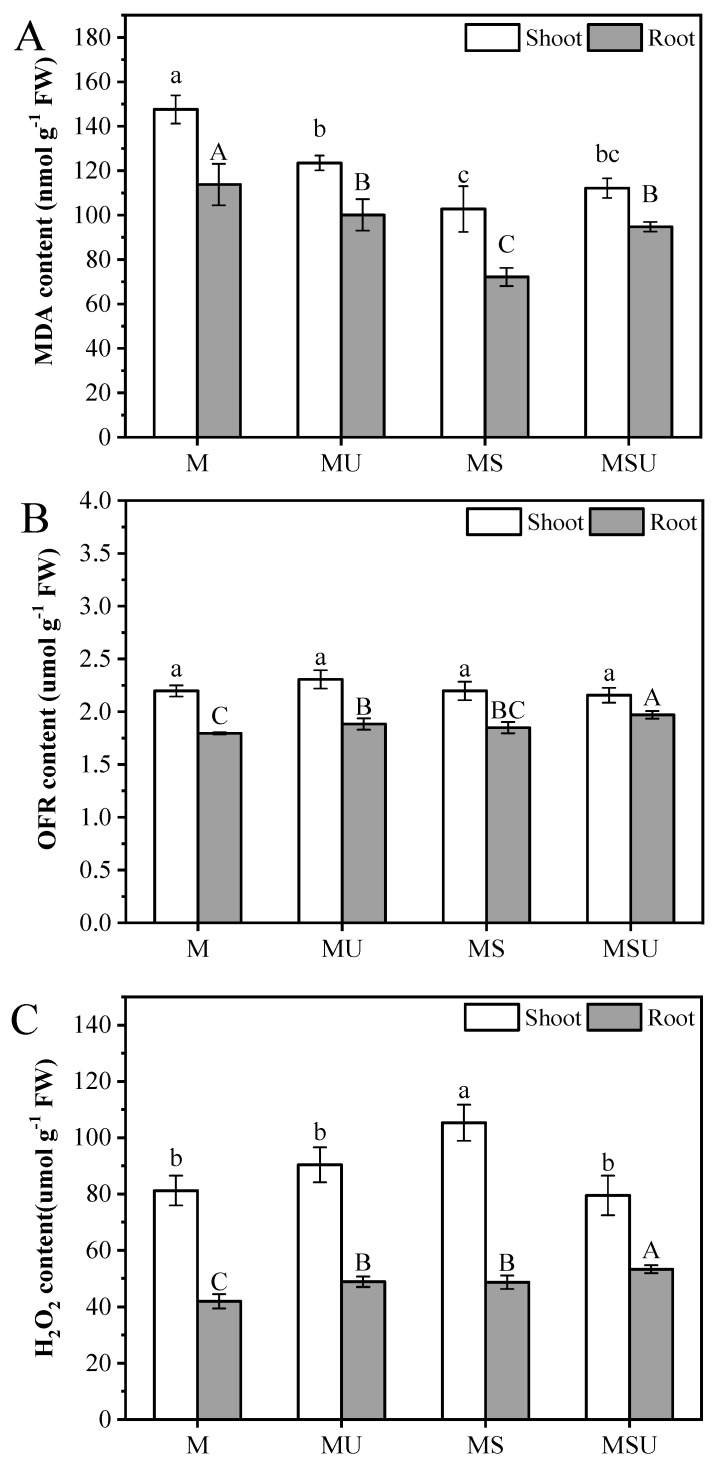
The contents of malondialdehyde (MDA) (**A**), oxygen free radical (OFR) (**B**), and hydrogen peroxide (H_2_O_2_) (**C**) in shoots and roots under the different rhizobium treatments. M: *M. sativa*, MU: *M. sativa* + urea, MS: *M. sativa* + *S. meliloti*, MSU: *M. sativa* + *S. meliloti* + urea. Bars are standard error (*n* = 3). Different letters stand for significant difference (*p* < 0.05) with Duncan’s post-test.

**Figure 3 ijerph-16-01955-f003:**
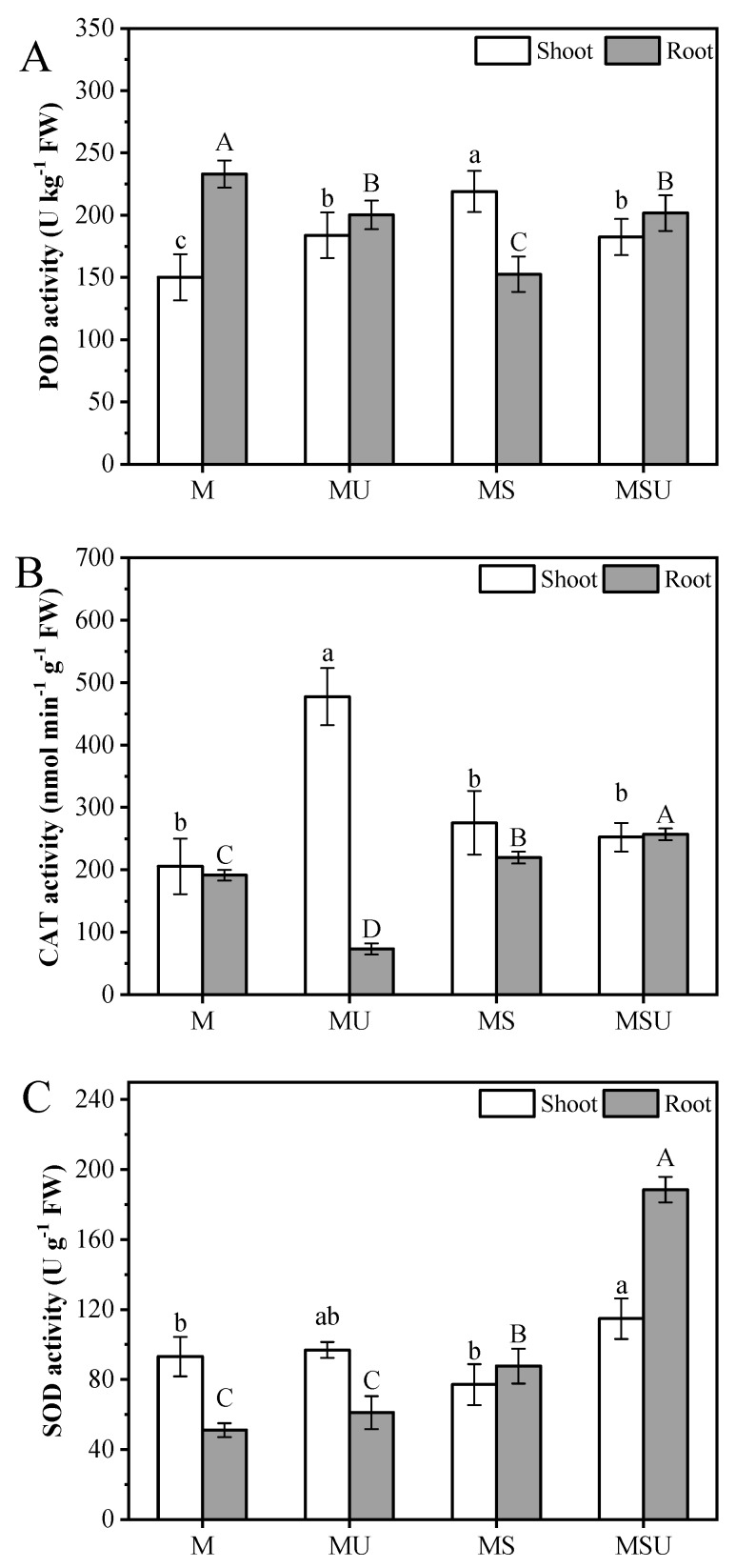
The antioxidant enzyme activities (SOD: superoxide dismutase (**C**); POD: peroxidase (**A**); CAT: catalase (**B**)) in shoots and roots under the different rhizobium treatments. M: *M. sativa*, MU: *M. sativa* + urea, MS: *M. sativa* + *S. meliloti*, MSU: *M. sativa* + *S. meliloti* + urea. Bars are standard error (*n* = 3). Different letters stand for significant difference (*p* < 0.05) with Duncan’s post-test.

**Figure 4 ijerph-16-01955-f004:**
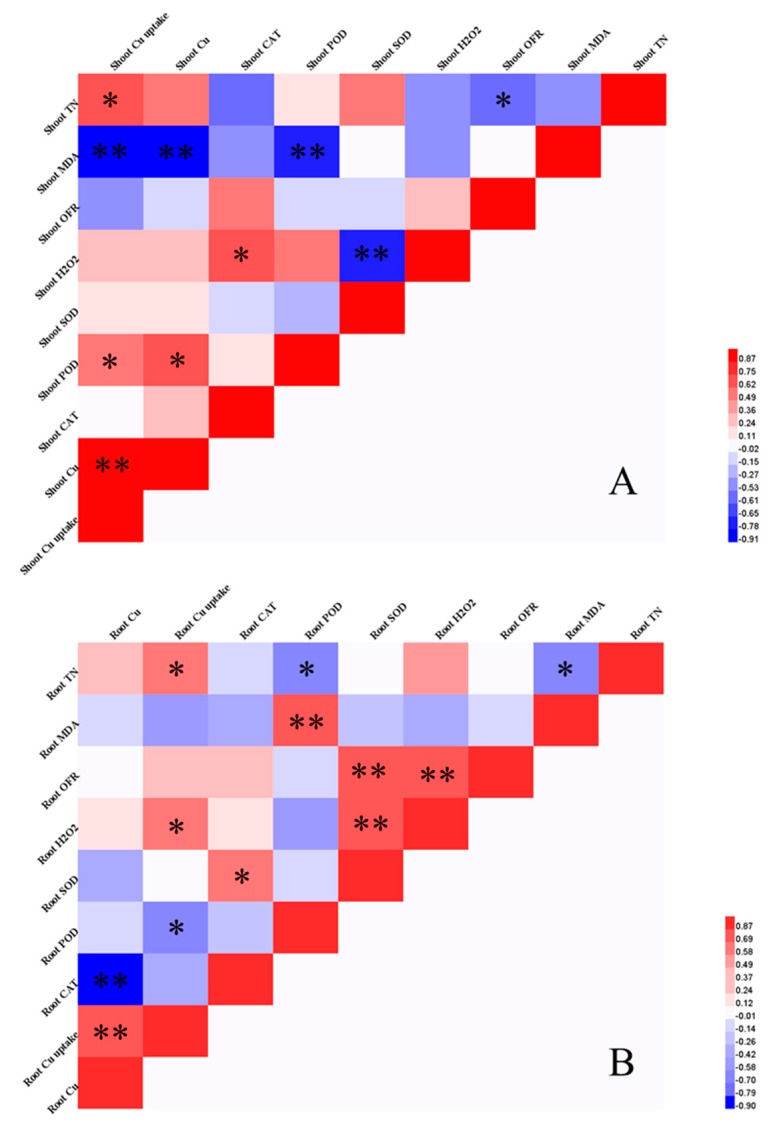
Heat maps of correlation of Cu content, Cu uptake, total N (TN), oxidative damage (MDA: malondialdehyde; OFR: oxygen free radical; H2O_2_: hydrogen peroxide), and antioxidant enzyme activities (SOD: superoxide dismutase; POD: peroxidase; CAT: catalase) in plant shoots (**A**) and roots (**B**) based on Pearson correlation coefficients. *, correlation is significant at *p*
*<* 0.05 (two-tailed); **, correlation is significant at *p* < 0.01. Strong positive correlation (red); strong negative correlation (blue).

**Table 1 ijerph-16-01955-t001:** Selected chemical properties of the soil sample used in the study.

Index	Value
pH	5.56 ± 0.10
SOM (g kg^−1^)	23.4 ± 2.13
AP (mg kg^−1^)	12.6 ± 0.34
AK (mg kg^−1^)	59.8 ± 1.75
TN (g kg^−1^)	1.10 ± 0.01
Pb (mg kg^−1^)	110 ± 9.8
Cu (mg kg^−1^)	688 ± 14.2
Zn (mg kg^−1^)	220 ± 9.6
Cd (mg kg^−1^)	0.60 ± 0.01
Available Cu (mg kg^−1^)	8.94 ± 0.28

SOM: soil organic matter; AP: available phosphorus; AK: available potassium; TN: total nitrogen. Values are the means ± standard error.

**Table 2 ijerph-16-01955-t002:** The soil properties after the experience.

Treatment	TN (g kg^−1^)		SOM (g kg^−1^)		Available Cu (mg kg^−1^)		Total Cu (mg kg^−1^)	
M	1.21 ± 0.01 c		24.6 ± 0.1 c		9.06 ± 0.34 b		646 ± 6.2 a	
MU	1.25 ± 0.01 b		24.6 ± 0.2 c		7.02 ± 0.24 c		604 ± 26.1 b	
MS	1.25 ± 0.01 b		26.1 ± 0.1 a		9.48 ± 0.41 b		608 ± 12.2 b	
MSU	1.30 ± 0.04 a		25.0 ± 0.3 b		10.5 ± 0.24 a		607 ± 9.0 b	
**Factors (Df)**	***F***	***p***	***F***	***p***	***F***	***p***	***F***	***p***
Inoculation (1)	13.0	**	68.2	***	114	***	3.99	0.08
Urea (1)	13.6	**	19.7	**	7.87	*	5.56	*
Inoculation * Urea (1)	0.007	0.79	20.5	**	69.7	***	5.28	0.05

TN: total nitrogen; SOM: soil organic matter. M: *M. sativa*, MU: *M. sativa* + urea, MS: *M. sativa* + *S. meliloti*, MSU: *M. sativa* + *S. meliloti* + urea. Values are the means ± standard errors (*n* = 3). Different letters stand for significant difference (*p* < 0.05) with Duncan’s post-test. Df, degrees of freedom. * *p* < 0.05, ** *p* < 0.01, *** *p* < 0.001.

**Table 3 ijerph-16-01955-t003:** The height/length, biomass, and chlorophyll content in plants under different treatments.

Treatments	Height/Length (cm)				Biomass (g Plant^−1^)				Chlorophyll Content (mg g^−1^ FW)	
	Shoot		Root		Shoot		Root			
M	33.2 ± 0.8 d		18.6 ± 0.8 d		0.29 ± 0.04 c		0.31 ± 0.02 c		1.69 ± 0.32 b	
MU	40.7 ± 1.1 c		21.4 ± 1.3 c		0.36 ± 0.01 b		0.38 ± 0.01 b		1.93 ± 0.18 ab	
MS	46.7 ± 1.6 b		35.1 ± 1.7 a		0.41 ± 0.01 ab		0.47 ± 0.01 a		2.26 ± 0.09 a	
MSU	51.5 ± 1.0 a		25.1 ± 1.0 b		0.46 ± 0.02 a		0.46 ± 0.01 a		1.86 ± 0.21 ab	
**Factors (Df)**	***F***	***p***	***F***	***p***	***F***	***p***	***F***	***p***	***F***	***p***
Inoculation (1)	333	***	188	***	47.6	***	202	***	0.99	0.35
Urea (1)	85.6	***	23.8	**	13.4	**	10.1	*	2.97	0.12
Inoculation * Urea (1)	4.00	0.08	74.5	**	0.42	0.54	23.9	***	6.56	*

M: *M. sativa*, MU: *M. sativa* + urea, MS: *M. sativa* + *S. meliloti*, MSU: *M. sativa* + *S. meliloti* + urea. FW: fresh weight. Values are the means ± standard errors (*n* = 3). Different letters stand for significant difference (*p* < 0.05) with Duncan’s post-test. Df, degrees of freedom. * *p* < 0.05, ** *p* < 0.01, *** *p* < 0.001.

**Table 4 ijerph-16-01955-t004:** The significance of the urea and rhizobium treatments (and interaction) on plant shoots.

Shoot	Inoculation		Urea		Inoculation * Urea	
	*F* Value	*p* Value	*F* Value	*p* Value	*F* Value	*p* Value
N content	29.7	**	1.32	0.28	11.6	**
OFR content	2.96	0.12	0.63	0.45	2.96	0.12
H_2_O_2_ content	0.30	0.60	0.02	0.90	28.2	**
MDA content	107	***	30.0	***	47.6	***
SOD activity	0.02	0.88	12.3	**	8.24	*
CAT activity	10.0	*	25.7	**	36.1	***
POD activity	11.7	**	0.02	0.89	12.7	**

Notes: Degrees of freedom (Df) = 1, within-group variance = 8. MDA: malondialdehyde; OFR: oxygen free radical; H_2_O_2_: hydrogen peroxide; SOD: superoxide dismutase; POD: peroxidase; CAT: catalase. * *p* < 0.05, ** *p* < 0.01, *** *p* < 0.001.

**Table 5 ijerph-16-01955-t005:** The significance of the urea and rhizobium treatments (and interaction) on plant roots.

Root	Inoculation		Urea		Inoculation * Urea	
	*F* Value	*p* Value	*F* Value	*p* Value	*F* Value	*p* Value
N content	3.03	0.12	0.00	0.992	12.2	**
OFR content	8.33	*	18.6	**	0.50	0.50
H_2_O_2_ content	21.3	**	22.6	**	0.92	0.37
MDA content	41.8	***	1.52	0.253	25.0	**
SOD activity	316	***	144	***	96.8	***
CAT activity	406	***	59.0	***	219	***
POD activity	28.4	**	1.20	0.30	30.6	**

Notes: Df (degrees of freedom) = 1, within-group variance = 8. MDA: malondialdehyde; OFR: oxygen free radical; H_2_O_2_: hydrogen peroxide; SOD: superoxide dismutase; POD: peroxidase; CAT: catalase. * *p* < 0.05, ** *p* < 0.01, *** *p* < 0.001.

**Table 6 ijerph-16-01955-t006:** The variation of Cu content, uptake, and transfer coefficient in plants.

Treatments	Cu Content (mg kg^−1^ DW)				Total Cu Uptake (ug Plant^−1^)				Bioconcentration Factor				Translocation Factor	
	Shoot		Root		Shoot		Root		Shoot		Root			
M	26.2 ± 1.6 c		59.5 ± 1.2 c		8.40 ± 1.3 c		18.6 ± 1.6 d		0.040 ± 0.002 c		0.092 ± 0.002 c		0.48 ± 0.01 b	
MU	35.0 ± 0.5 ab		91.8 ± 1.4 a		11.9 ± 0.6 b		34.9 ± 1.1 a		0.058 ± 0.001 b		0.152 ± 0.002 a		0.36 ± 0.02 c	
MS	36.6 ± 0.7 ab		66.5 ± 3.0 b		14.2 ± 0.8 a		31.3 ± 1.7 b		0.060 ± 0.001 ab		0.109 ± 0.005 b		0.52 ± 0.01 ab	
MSU	38.2 ± 2.1 a		61.5 ± 3.8 c		15.9 ± 1.0 a		28.1 ± 1.7 c		0.063 ± 0.004 a		0.101 ± 0.006 b		0.57 ± 0.06 a	
**Factors (Df)**	***F***	***p***	***F***	***p***	***F***	***p***	***F***	***p***	***F***	***p***	***F***	***p***	***F***	***p***
Inoculation (1)	91.1	***	59.8	***	76.8	***	10.8	*	95.8	***	46.1	***	42.7	***
Urea (1)	50.1	***	82.2	***	21.3	**	52.1	***	69.8	***	111	***	3.86	0.08
Inoculation * Urea (1)	20.3	**	153	***	2.90	0.13	117	***	31.9	***	188	***	18.8	**

M: *M. sativa*, MU: *M. sativa* + urea, MS: *M. sativa* + *S. meliloti*, MSU: *M. sativa* + *S. meliloti* + urea. DW: dry weight. The bioconcentration factor indicates the ability of plants to concentrate heavy metals into their tissues from the surrounding soil environment, “Bioconcentration factor = concentration of metal in plant tissues / concentration of metal in soil”. The transport ability of Cu from roots to shoots in the plant is given by Transfer factor, “Translocation factor = shoot Cu content / root Cu content”. Values are the means ± standard errors (*n* = 3). Different letters stand for significant difference (*p* < 0.05) with Duncan’s post-test. Df, degrees of freedom. * *p* < 0.05, ** *p* < 0.01, *** *p* < 0.001.

**Table 7 ijerph-16-01955-t007:** Correlation analysis between properties of soil and plant.

Shoot	Cu Uptake	CAT	POD	SOD	OFR	MDA	H_2_O_2_
Soil TN	0.86 **	0.15	0.36	0.32	−0.19	−0.72 **	0.15
SOM	0.54	−0.19	0.79 **	−0.49	−0.22	−0.63 *	0.47
Available Cu	0.46	−0.78 **	0.06	0.17	−0.57	−0.18	−0.43
Total Cu	−0.70 *	−0.42	−0.52	0.05	0.03	0.69 *	−0.54
Root							
Soil TN	0.48	0.33	−0.27	0.78 **	0.77 **	−0.28	0.79 **
SOM	0.31	0.48	−0.81 **	0.19	0.016	−0.87 **	0.27
Available Cu	−0.39	0.97 **	−0.12	0.69 *	0.33	−0.22	0.26
Total Cu	−0.70 *	0.08	0.56	−0.33	−0.72 **	0.42	−0.61 *

MDA: malondialdehyde; OFR: oxygen free radical; H_2_O_2_: hydrogen peroxide and antioxidant enzyme activities (SOD: superoxide dismutase; POD: peroxidase; CAT: catalase) in plant shoots based on Pearson correlation coefficients. *, correlation is significant at *p* < 0.05 (two-tailed); **, correlation is significant at *p* < 0.01 (two-tailed). SOM: soil organic matter; TN: total N.

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
