# Peer review of "Impact of Urea Addition and Rhizobium Inoculation on Plant Resistance in Metal Contaminated Soil"

_ijerph, 2019, doi:10.3390/ijerph16111955_

Round 1

Reviewer 1 Report

Comments on “Deciphering the Role of Nitrogen on the Resistance of Legume-Rhizobium Symbiotic System in Copper Contaminated Soil”

Major revision

Plant-microbe interactions have been proven to be important in soil remediation. Of which, the microorganisms may enhance plant growth and thus increase metal accumulation. I don’t think the main content of this paper is “the role of N on……”, it seems that the objective is to evaluate the roles of rhizobium and N on plant growth and Cu accumulation. So, the title should be revised and all the conclusions in this manuscript should be highlighted by a same logic throughout the paper. Other suggestions for improving this paper are given below.

Major gaps:

1.       Line 138. Why used sterilized water?

2.       Line 188. The available Cu is given in Table 2, why it is not given in Table 1? Moreover, the authors should have more descriptions on the data shown in Table 2, e.g., why TN and SOM increased as compared to that in Table 1? The soil data is very important.

3.       Lines 199-200. The authors should explain why inhibitory effects occurred in DISSCUSSION section. Similar confuse is also found in Table 4, why the shoots and roots show different trend, i.e., the highest data is found in MU groups.

4.       Line 264. How do you check the influence of rhizobacteria on Cu uptake?

5.       Lines 316-319. Have you determined the enzyme activities of the rhizobacteria you used?

Minor mistakes:

1.         Lines 57-49. Pls remove the last sentence as it doesn’t make sense in this paper.

2.         Line 92. It is unnecessary to define the abbreviation of Medicago sativa. You can use M. sativa for the 2nd appearance directly.

3.         Line 132. Pls rewrite the sentence “as the Sobrino-Plata et al. [39] method”.

4.         Line 146, 151 and 161. Pls place “an” with “a”.

5.         Lines 153-154. Pls rewrite the last sentence.

6.         Line 158. Pls keep a blank space between 15000 and g, and check the similar mistakes throughout the paper.

7.         Line 183. Pls check p.

8.         Line 338. Pls replace “suggest” with “suggesting”.

9.         Line 395. Medicago sativa, pls use the abbreviated name.

Author Response

Point-to-point response to reviewer comments

Response to reviewer #1:

General comments: Plant-microbe interactions have been proven to be important in soil remediation. Of which, the microorganisms may enhance plant growth and thus increase metal accumulation. I don’t think the main content of this paper is “the role of N on……”, it seems that the objective is to evaluate the roles of rhizobium and N on plant growth and Cu accumulation. So, the title should be revised and all the conclusions in this manuscript should be highlighted by a same logic throughout the paper. Other suggestions for improving this paper are given below.

Response: We very appreciate your positive comments and constructive suggestions for our manuscript. Your comments and recommendations have been considered seriously and most addressed in our revised manuscript. According to your comments, the title “Deciphering the Role of Nitrogen on the Resistance of Legume-Rhizobium Symbiotic System in Copper Contaminated Soil” has been revised to “Impact of Urea Addition and Rhizobium Inoculation on Plant Resistance in Metal Contaminated Soil” and some related parts have been modified in the revised manuscript. Detailed responses are as below.

Major comments

Question 1:  Line 138. Why used sterilized water?

Response: Thanks for your careful review, we are sorry for the error description in the handling method that caused confusing. The “sterilized water” have been amended to “distilled water” in the revised manuscript.

The harvested shoots and roots were washed with distilled water carefully, after samples were dried at 65 °C for 48 h, the plant samples were then separated into two portions. (Lines 141-142)

Question 2: Line 188. The available Cu is given in Table 2, why it is not given in Table 1? Moreover, the authors should have more descriptions on the data shown in Table 2, e.g., why TN and SOM increased as compared to that in Table 1? The soil data is very important.

Response: Thank you for pointing out this issue. The data of available Cu have been added in the Table 1. About the issue of why TN and SOM increased as compared to that in Table 1 you mentioned, we have added more explanations on the variation of original data in the discussion. The corresponding contents can be seen in the revised manuscript (Lines 328-335).

In our study, the result showed that the content of N and SOM increased after alfalfa planting in metals contaminated soil. This result confirmed that the presence of root nodules and the assimilation of free N thus increasing the N content in soil [42]. And the reason for the increased of SOM content could be that root exudates and other root-borne organic substance released into the rhizosphere during the plant growth, as well as root hairs and fine roots sloughed by root elongation [43]. Besides, the legume has been proved that it is loose growth and rots easily [44]. In terms of available Cu in soil, it was also increased after planting then suggesting the exist of root was effective to remediation of contaminated soil. (Lines 328-335)

References

[42] Van Kessel, C.; Hartley, C. Agricultural management of grain legumes: has it led to an increase in nitrogen fixation? Field Crops Res. 2000, 65(2), 165-181.

[43] Fred, E.B.; Baldwin, I.L.; McCoy, E. Root Nodule Bacteria and Leguminous Plants. University of Wisconsin Press: Madison, USA, 1932; 343.

[44] Kuzyakov, Y.; Domanski, G. Carbon input by plants into the soil. Review. J. Plant Nutr. Soil Sci. 2000, 163(4), 421-431.

Question 3: Lines 199-200. The authors should explain why inhibitory effects occurred in DISSCUSSION section. Similar confuse is also found in Table 4, why the shoots and roots show different trend, i.e., the highest data is found in MU groups.

Response: According to your comments, we further modified the result and discussion sections. About the issue of inhibitory effects that you mentioned, we have corrected the sentence in the revised edition. In our manuscript, we have made an explanation “previous experimental results showed a decrease in the biomass of soybean nodules when grown with N inferring that N addition reduces rhizobial performance [51] (Lines 355-357)” and “is likely due to reduced root colonization from N fertilization [53] (Lines 363-364)”. About the issue of the highest data is found in MU groups, we had added a more detailed explanation in discussion section “The highest Cu uptake in the roots of urea treated plants (Table 2) can be explained by that NH4+ could change the subcellular distribution (cell wall and vacuole could bind and sequestration metals thus limiting their translocation to shoot) [54] and chemical forms of metal (for example, metals integrated with pectates and protein generally have a lower migration) [54, 55] (Lines 364-368)”. In terms of lower total Cu uptake of plant shoots than inoculation groups, the reason was that inoculation have a better potential to enhance the biomass accumulation. The corresponding contents can be seen in the revised manuscript (Lines 209-210, 355-357, 362-364, 364-368).

In MSU, the root length was shorter relative to the individual inoculation treatment (MS). (Lines 209-210)

In agreement, previous experimental results showed a decrease in the biomass of soybean nodules when grown with N inferring that N addition reduces rhizobial performance [51]. (Lines 355-357)

The lower Cu uptake observed in plant roots treated with both urea and rhizobium inoculation relative to inoculation alone, is likely due to reduced root colonization from N fertilization [53]. (Lines 362-364)

The highest Cu uptake in the roots of urea treated plants (Table 2) can be explained by that NH4+ could change the subcellular distribution (cell wall and vacuole could bind and sequestration metals thus limiting their translocation to shoot) [54] and chemical forms of metal (for example, metals integrated with pectates and protein generally have a lower migration) [54, 55]. (Lines 364-368)

References

[55] Lai, H.Y. Subcellular distribution and chemical forms of cadmium in Impatiens walleriana in relation to its phytoextraction potential. Chemosphere 2015, 138, 370-376.

Question 4:   Line 264. How do you check the influence of rhizobacteria on Cu uptake?

Response: In your mentioned line 264 we referred to the divergent effects of environmental variables on Cu uptake, the relaimpo package in R 3.5.2 was used to calculates the relative importance of predictor variables for the linear model (Groemping and Matthias, 2018). We used this package to identify the most important variables explaining the variation in Cu uptake in plants. Rhizobacteria could affect the environmental variables, such as soil properties, and plant physiological and biochemical properties. The environmental variables could influence the Cu uptake, for example, the content of soil N and organic matter increased with rhizobium inoculation accompanying with accumulation of Cu uptake in plant (see Tables 2 and 4). Previous study also reported that the nitrogen addition improved the phytoextraction efficiency in Cu contaminated soil (Fang et al., 2017). The increase of soil organic matter could stimulate the metals uptake in plants. Besides, metal uptake by plants dependent on high environmental metal concentrations (Ignacio Barquero et al., 2019).

Reference:

Groemping, U.; Matthias, L. 2018. Package ‘relaimpo’.

Fang, L.C.; Wang, M.K.; Cai, L.; Cang, L. Deciphering biodegradable chelant-enhanced phytoremediation through microbes and nitrogen transformation in contaminated soils. Environ. Sci. Pollut. Res. 2017, 24(17), 1-10.

Ignacio Barquero, J.; Rojas, S.; María Esbrí, J.; M. García-Noguero, E.; Higueras, P. Factors influencing mercury uptake by leaves of stone pine (Pinus pinea L.) in Almadén (Central Spain). Environ. Sci. Pollut. Res. 2019, 26(4), 3129-3137.

Question 5:  Lines 316-319. Have you determined the enzyme activities of the rhizobacteria you used?

Response: About the issue you mentioned, previous research reported that the strain S. meliloti CCNWSX0020 had the high level of ACC deaminase activity (Kong et al., 2015; Ma et al., 2004). Also, we have examined that the strain S. meliloti CCNWSX0020 was able to endure higher Cu concentrations (see Figure S1).

The rhizobium S. meliloti was able to endure higher concentrations of Cu in this study (Fig. S1). (Lines 113-114)

Reference:

Kong, Z.Y.; Glick, B.R.; Duan, J.; Ding, S.L.; Tian, J.; McConkey, B.J.; Wei, G.H. Effects of 1-aminocyclopropane-1-carboxylate (ACC) deaminase-overproducing Sinorhizobium meliloti on plant growth and copper tolerance of Medicago lupulina. Plant Soil 2015, 391(1), 383-398.

Ma, W.; Charles, T.C.; Glick. B.R. Expression of an exogenous 1-aminocyclopropane-1-carboxylate deaminase gene in Sinorhizobium meliloti increases its ability to nodulate alfalfa. Appl. Environ. Microbiol. 2004, 70, 5891-589.

Minor comments

Question 1:  Lines 57-49. Pls remove the last sentence as it doesn’t make sense in this paper.

Response: According to your comment, the sentence has been deleted.

Question 2:  Line 92. It is unnecessary to define the abbreviation of Medicago sativa. You can use M. sativa for the 2nd appearance directly.

Response: Thanks for your careful review. According to your suggestions, the word has been modified in the revised manuscript.

M. sativa seeds (provided by Beijing Rytway Ecotechnology Co., LTD, China) were surface-sterilized by 75% v/v ethanol for 3 minutes, followed by 10 minutes in 20% v/v NaClO (containing 8% available chlorine). (Lines 95-97)

Question 3: Line 132. Pls rewrite the sentence “as the Sobrino-Plata et al. [39] method”.

Response: According to your comment, the sentence has been rewritten.

The total chlorophyll content in leaves was determined as the method of Sobrino-Plata et al. [40]. (Lines 134-135)

Question 4: Line 146, 151 and 161. Pls place “an” with “a”.

Response: According to your comment, these sentences have been edited again in the revised manuscript (Lines 148-150, 154-155, 162-165).

The MDA content was measured using a MDA reagent kit (Suzhou Comin Biotechnology Co., Ltd. Suzhou, China). (Lines 148-150)

Hydrogen peroxide (H2O2) was measured using a H2O2 reagent kit (Suzhou Comin Biotechnology Co., Ltd. Suzhou, China). (Lines 154-155)

The activity of total superoxide dismutase (SOD, EC 1.15.1.1) was assayed by determining the ability of inhibiting photochemical reduction of nitroblue tetrazolium (NBT) according to the instructions of a SOD reagent kit (Suzhou Comin Biotechnology Co., Ltd. Suzhou, China). (Lines 162-165)

Question 5:  Lines 153-154. Please rewrite the last sentence.

Response: Thanks for your comment, the sentence has been rewritten in the revised manuscript.

The above operations all according to the manufacturer's instructions. (Lines 156-157)

 Question 6: Line 158. Pls keep a blank space between 15000 and g, and check the similar mistakes throughout the paper.

Response: According to your comment, we have modified the similar mistakes throughout the manuscript.

The homogenate was centrifuged at 15000 g for 15 min at 4 °C and the supernatant was used for assaying antioxidant enzyme activities [37]. (Lines 161-162)

Question 7: Line 183. Pls check p.

Response: According to your comment, "P" has been replaced by "p".

The total N content significantly increased by 3.30%, 3.30%, and 7.44% in MU, MS, and MSU relative to M, respectively (P < 0.05). (Lines 186-187)

 Question 8: Line 338. Pls replace “suggest” with “suggesting”.

Response: Thanks for your nice suggestion, “suggest” has been modified to “suggesting” in the revised manuscript.

Furthermore, higher shoot translocation and bioconcentration factors observed in inoculated plants compared with urea treated plants (Table 4), suggesting that inoculation facilitated Cu phytoextraction [23]. (Lines 371-373)

Question 9:  Line 395. Medicago sativa, pls use the abbreviated name.

Response: Thanks for your careful review. We have modified our sentences in the revised manuscript.

This study investigates the effects of both urea or/and rhizobial symbiosis on the Cu resistance of the legume species M. sativa in copper-contaminated soil. (Lines 429-430)

Reviewer 2 Report

The present study provides valuable information about the growth and Cu uptake of Medicago sativa in a Cu polluted soil as affected by the presence of Sinorhizobium melitoti and N-fertilization with urea. The text is well structured and the methods are sound, but there are some aspects of the manuscript that need to be revised. Below, I provide some comments that might be useful to the authors.

Introduction section
This section is well organized. The first paragraph talks about Cu pollution; the second paragraph focuses on the response of plants to the addition of N-fertilizers in Cu contaminated soils. The third paragraph introduces the positive effect of rhizobium on Cu resistance and Cu uptake by plants. Unfortunately, the authors do not introduce properly the possible interactions between rhizobium and N availability for metal resistance (only a sentence in lines 74-75). One or two additional sentences regarding the basis of this beneficial interaction would significantly improve the quality of the introductory section.

Material and Methods section
As indicated before, materials are adequate and the methods seem sound. I have only one concern regarding the statistical analysis. The R package relaimpo (not relaimpog) calculates the relative importance of regressors in a linear model. In this study, according to the chosen experimental design, the regressors are U, S and U:S. However, the authors apply the relaimpo package to a dataset including all the response variables they have chosen, as if they were inferring the relative contribution of a set of predictor variables measured in a field survey. In my opinion this difficults the interpretation of the results. I recommend to apply the relaimpo package to calculate the relative importance of the anova model regressors. The study will also benefit from a proper discussion of the obtained results.

Results section
Although in the M&M section the authors referred to an ANOVA analysis, the results of such analysis are never reported. The significance of the urea and rhizobium treatments (and their interaction) must be properly reported indicating the F statistic, the degrees of freedom, the associated probability and the sample size. The authors should explore the possibility of including the part of the plant (shoot / root) as an additional factor in their Anova model. The authors can keep separated the analysis of the results obtained from the shoots and roots, but in my opinion the interpretation of the results would benefit from including this additional factor in the model.

Discussion section
This section is very complete and the authors demostrate a detailed knowledge of the literature. In general, the discussion is quite accurate, but as I mentioned in my comments to the Results section, this part of the manuscript would benefit from a detailed discussion of the Aova results, explaining the results from the point of view of main versus interaction effects.  

General comment: the English of the manuscript needs to be revised. Some words are not properly used or, in some cases, some sentences are not clear, e.g., the use of "prolific" instead of "frequent" in line 35, the use of "concern" in lines 42-43, or "identified to conserve" in line 67. Small mistakes like these are scattered all throughout the text, so a careful revision is needed.

Author Response

Point-to-point response to reviewer comments

Response to reviewer #2:

General comment: the English of the manuscript needs to be revised. Some words are not properly used or, in some cases, some sentences are not clear, e.g., the use of "prolific" instead of "frequent" in line 35, the use of "concern" in lines 42-43, or "identified to conserve" in line 67. Small mistakes like these are scattered all throughout the text, so a careful revision is needed.

Response: We appreciate the reviewer’s careful review. We again carefully checked the grammar and word of the manuscript, and requested Editage (www.editage.cn) for linguistic check of manuscript once again. Meanwhile, the language editing certificate of this manuscript has been enclosed. Additionally, your comments and recommendations have been considered seriously and addressed in our revised manuscript (Lines 34-35, 42-43, 66-67).

Copper (Cu) contamination is one of the most widespread and damaging types of heavy metal pollution in terrestrial ecosystems [3, 4]. (Lines 34-35)

The application of phytoremediation for contamination treatment has raised general concerns because it is both economical and environmentally friendly [6, 8]. (Lines 42-43)

The ion-selective ATPase pumps have also been observed to conserve the metal transfer system in rhizobium [25-27]. (Lines 66-67)

Comments 1:

Introduction section
This section is well organized. The first paragraph talks about Cu pollution; the second paragraph focuses on the response of plants to the addition of N-fertilizers in Cu contaminated soils. The third paragraph introduces the positive effect of rhizobium on Cu resistance and Cu uptake by plants.
Unfortunately, the authors do not introduce properly the possible interactions between rhizobium and N availability for metal resistance (only a sentence in lines 74-75). One or two additional sentences regarding the basis of this beneficial interaction would significantly improve the quality of the introductory section.
Response:
We very appreciate the reviewer’s positive comments on the organization and description of our introduction section. We also agree with you mentioned that additional description of possible interactions between rhizobium and N availability for metal resistance. According to your comments and suggestions, we further modified this section. The corresponding contents can be seen in the revised manuscript (Lines 60-61, 73-75).

The symbiotic nitrogen fixation also could promote plant growth thus increasing plant biomass under metal contaminated environment [11]. (Lines 60-61)

In previous study, since without external N supply, N content increased with rhizobium inoculation accompanying by promoting related gene expression abundances of antioxidants enzymes [34]. (Lines 73-75)

Comments 2:

Material and Methods section
As indicated before, materials are adequate and the methods seem sound. I have only one concern regarding the statistical analysis. The R package relaimpo (not relaimpog) calculates the relative importance of regressors in a linear model. In this study, according to the chosen experimental design, the regressors are U, S and U:S. However, the authors apply the relaimpo package to a dataset including all the response variables they have chosen, as if they were inferring the relative contribution of a set of predictor variables measured in a field survey. In my opinion this difficults the interpretation of the results. I recommend to apply the relaimpo package to calculate the relative importance of the anova model regressors. The study will also benefit from a proper discussion of the obtained results.

Response: Thanks for your nice suggestion. According to your suggestion, we choose the significant variables (based on an ANOVA) to identify the most important variables explaining the relative importance of the variation in Cu uptake in shoots and roots using the relaimpo package in R 3.5.2. The related contents have been rewritten in the revised manuscript (Lines 173-176, 286-294).

To identify the most important variables explaining the variation in Cu uptake in shoots and roots, we determined measures of relative importance of predictor variables (based on an ANOVA) using the relaimpo package in R 3.5.2. (Lines 173-176)

The following significant environmental variables explained 94.24% of the variation in shoot Cu uptake: soil TN, available Cu, organic matter, and plant N, MDA, and POD (Figure 4A). Of these variables, plant MDA content was the most significant of all variables, explaining 32.6% of the variation in shoot Cu uptake. Soil TN and plant N contents were also key variables, explaining 27.0% and 11.5% of the variation in shoot Cu uptake, respectively. These environmental variables explained 98.92% of the variation of Cu uptake in roots (Figure 4B). As with shoot Cu uptake, plant MDA content was also the most important variable influencing Cu uptake in roots, accounting for 16.2% of the variation. CAT activity and H2O2 content in roots were also key variables influencing root Cu uptake, accounting for 14.2% and 14.0% of the variation, respectively. (Lines 286-294)

Comments 3:

Results section
Although in the M&M section the authors referred to an ANOVA analysis, the results of such analysis are never reported. The significance of the urea and rhizobium treatments (and their interaction) must be properly reported indicating the F statistic, the degrees of freedom, the associated probability and the sample size. The authors should explore the possibility of including the part of the plant (shoot / root) as an additional factor in their Anova model. The authors can keep separated the analysis of the results obtained from the shoots and roots, but in my opinion the interpretation of the results would benefit from including this additional factor in the model.
Response: According to your suggestions, two-way univariate analysis of variance (ANOVA) was carried out to analysis the significance of the urea and rhizobium treatments (and their interaction). About the suggestion you mentioned that explore the possibility of including the part of the plant (shoot / root) as an additional factor in their ANOVA model was drawn seriously attention by us, the result of analysis exactly reflected the overall situation about the effect of different treatments on plant properties. For example, the result confirmed that the related oxidative stresses were the lowest with individual rhizobium inoculation than other treatments. But separated the analysis of the results obtained from the shoots and roots showed that antioxidant enzymes present different responses in shoots and roots (see Figure 3), this has also been reported in previous study (Chen et al., 2018). Thus we think that separated analysis would be better. You can find it in the revised manuscript (Lines 171-173, 187-188, 190-193, 206-207, 386-388, 404-407, 410-413).

Two-way univariate analysis of variance (ANOVA) was conducted to analyze the effects of urea and rhizobium inoculation on properties of soil and plant with SPSS 20.0 (SPSS, Chicago, Illinois USA). Duncan's post-test (P < 0.05) was used for post hoc investigation. (Lines 171-173)

In terms of SOM, urea and inoculation also had an influence, and two factors had an interactive effect (Table 2). (Lines 187-188)

The available Cu in soil enhanced by 4.64% and 15.8% in MS and MSU compared to M, respectively, but significantly decreased by 22.5% after urea addition alone (the interaction of factors was significant (Table 2)). Besides, only the urea treatment affected the total Cu content (F = 5.559, P = 0.046). (Lines 190-193)

And the urea and rhizobium inoculation had a significant interaction influence on root length, root biomass, and chlorophyll content (Table S1, S2). (Lines 206-207)

Plants with both urea and rhizobium had the highest value of OFR in their roots, and there was no interaction between urea and rhizobium inoculation (Table S2), possibly suggesting that external N supply was harmful to the inoculated plant roots. (Lines 386-388)

This observation suggests that inoculation alone triggers the rise of POD activity in plant shoots. Rhizobium inoculation had no effect on shoot SOD activity (Table S1). The above indicated that urea and rhizobium inoculation have a different resistance response to Cu stress in plant shoots. (Lines 404-407)

Meanwhile, our result showed that both urea and rhizobium inoculation had a significant effect on the activity of SOD and CAT in roots (Table S2). Interestingly, compared with the control group, inoculation significantly enhanced the activities of SOD and CAT, indicating that both enzymes play a central role in improving plant root resistance. (Lines 410-413)

Reference:

Chen, J.; Liu, Y.Q.; Yan, X.W.; Wei, G.H.; Zhang, J.H.; Fang, L.C. Rhizobium inoculation enhances copper tolerance by affecting copper uptake and regulating the ascorbate-glutathione cycle and phytochelatin biosynthesis-related gene expression in Medicago sativa seedlings. Ecotoxicol. Environ. Saf. 2018, 162, 312-323.

Comments 4:
Discussion section
This section is very complete and the authors demostrate a detailed knowledge of the literature. In general, the discussion is quite accurate, but as I mentioned in my comments to the Results section, this part of the manuscript would benefit from a detailed discussion of the Anova results, explaining the results from the point of view of main versus interaction effects.
Response: Thank you for your positive comments and nice suggestions. According to your comments and suggestions, we further modified the discussion section. The detailed corresponding contents can be seen in the revised manuscript (Lines 353-355, 386-388, 400-401, 405-407).

As ANOVA analysis showed that urea and rhizobium inoculation had a significant interaction in root length and biomass. (Lines 353-355)

Plants with both urea and rhizobium had the highest value of OFR in their roots, and there was no interaction between urea and rhizobium inoculation (Table S2), possibly suggesting that external N supply was harmful to the inoculated plant roots. (Lines 386-388)

Urea application enhanced enzyme activity, including CAT and SOD, suggesting that both enzymes are vital constituents in the defence mechanism of plant shoots (Table S1). (Lines 400-401)

Rhizobium inoculation had no effect on shoot SOD activity (Table S1). The above indicated that urea and rhizobium inoculation have a different resistance response to Cu stress in plant shoots. (Lines 405-407)

Reviewer 3 Report

The paper by Shen et al. has reached some interesting data about the role of the nitrogen and rhizobium on Medicago sativa  heavy metals tolerance. However in my opinion the paper has a few weak point. The paper lacks important methodological information. The title and aim of the study is improperly formulated. However, the paper is a potentially good study and after a few corrections can be published in the IJERPH.

Firstly, since the authors used soil collected from natural site which is contaminated not only by Cu but also by Pb (and probably by others heavy metals) I suggest to reconsider the title and the major aim of the paper – not influence of only Cu but impact of multi-heavy metals stress. The table 1 must be completed by concentration of other heavy metals –especially content of Cd.

p3. l. 100 – the urea concentration used must be given  

p3. l. 119 – I cannot find a data of CaCl2-extractabel Cu concentration

p4. l.168 – I suggest to perform 2-way ANOVA (1 factor = urea; 2 factor = S. meliloti) and please show interaction between these two factors and level of variability.

p5. l.188 – the table captions should be improve - for example: (…) properties in soil after experiment

Figure 1.; Table 4 (and others) please add FW or DW in description of axis (mg kg-1 DW ?)

I found unjustified statement p. 13 l. 316-317 (lack of data which can be supported this suggestion – too speculative).

Author Response

Point-to-point response to reviewer comments

Response to reviewer #3:

The paper by Shen et al. has reached some interesting data about the role of the nitrogen and rhizobium on Medicago sativa heavy metals tolerance. However in my opinion the paper has a few weak point. The paper lacks important methodological information. The title and aim of the study is improperly formulated. However, the paper is a potentially good study and after a few corrections can be published in the IJERPH.

Question 1: Firstly, since the authors used soil collected from natural site which is contaminated not only by Cu but also by Pb (and probably by others heavy metals) I suggest to reconsider the title and the major aim of the paper – not influence of only Cu but impact of multi-heavy metals stress. The table 1 must be completed by concentration of other heavy metals –especially content of Cd.

Response: According to your nice suggestions, the Cd content in the soil we collected was 0.60 mg kg-1, and this has been added in the Table 1. We agree with your opinion that the soil we used was polluted by multi-heavy metals, thus we have revised the title “Deciphering the Role of Nitrogen on the Resistance of Legume-Rhizobium Symbiotic System in Copper Contaminated Soil” to “Impact of Urea Addition and Rhizobium Inoculation on Plant Resistance in Metal Contaminated Soil” and some related parts have been modified in the revised manuscript. Detailed responses are as below.

Question 2: p3. l. 100 – the urea concentration used must be given

Response: Thanks for your careful review. We are sorry for this description may not be specific enough, the used urea concentration was 139.2 mg kg-1. The detailed modifications have been added in the Materials and methods section (Lines 117-118).

On the 60th day of plant growth, urea solution was applied to the soil surface with concentration of 5 mmol kg-1 (139.2 mg kg-1). (Lines 117-118)

Question 3: p3. l. 119 – I cannot find a data of CaCl2-extractable Cu concentration

Response: The data of CaCl2-extractable Cu concentration some contents have been added in the Table 1.

Question 4: p4. l.168 – I suggest to perform 2-way ANOVA (1 factor = urea; 2 factor = S. meliloti) and please show interaction between these two factors and level of variability.

Response: According to your suggestions, two-way univariate analysis of variance (ANOVA) was carried out to analysis the interaction between urea and rhizobium these two factors and level of variability. We have revised our manuscript and attached the related information in supplementary material (see Tables 2, S1, and S2). You can find it in the revised manuscript (Lines 171-173, 187-188, 190-193, 206-207, 386-388, 404-407, 410-413).

Two-way univariate analysis of variance (ANOVA) was conducted to analyze the effects of urea and rhizobium inoculation on properties of soil and plant with SPSS 20.0 (SPSS, Chicago, Illinois USA). Duncan's post-test (P < 0.05) was used for post hoc investigation. (Lines 171-173)

In terms of SOM, urea and inoculation also had an influence, and two factors had an interactive effect (Table 2). (Lines 187-188)

The available Cu in soil enhanced by 4.64% and 15.8% in MS and MSU compared to M, respectively, but significantly decreased by 22.5% after urea addition alone (the interaction of factors was significant (Table 2)). Besides, only the urea treatment affected the total Cu content (F = 5.559, P = 0.046). (Lines 190-193)

And the urea and rhizobium inoculation had a significant interaction influence on root length, root biomass, and chlorophyll content (Table S1, S2). (Lines 206-207)

Plants with both urea and rhizobium had the highest value of OFR in their roots, and there was no interaction between urea and rhizobium inoculation (Table S2), possibly suggesting that external N supply was harmful to the inoculated plant roots. (Lines 386-388)

This observation suggests that inoculation alone triggers the rise of POD activity in plant shoots. Rhizobium inoculation had no effect on shoot SOD activity (Table S1). The above indicated that urea and rhizobium inoculation have a different resistance response to Cu stress in plant shoots. (Lines 404-407)

Meanwhile, our result showed that both urea and rhizobium inoculation had a significant effect on the activity of SOD and CAT in roots (Table S2). Interestingly, compared with the control group, inoculation significantly enhanced the activities of SOD and CAT, indicating that both enzymes play a central role in improving plant root resistance. (Lines 410-413)

Question 5: p5. l.188 – the table captions should be improve - for example: (…) properties in soil after experiment

Response: Thanks for your comments and suggestions, the table caption has been modified in the revised manuscript (see Table 2).

Question 6: Figure 1.; Table 4 (and others) please add FW or DW in description of axis (mg kg-1 DW?)

Response: According to your comments, these Figures and Tables have been modified in the revised manuscript (see Tables 3 and 4, Figures 1, 2, and 3).

Question 7: I found unjustified statement p. 13 l. 316-317 (lack of data which can be supported this suggestion – too speculative).

Response: We agree with your opinion that there was lack of data which can be supported this suggestion. In fact, the engineered S. meliloti strain indeed has the high level of ACC deaminase activity (Kong et al., 2015; Ma et al., 2004). Corresponding modifications can be seen in the revised manuscript (Lines 346-347).

These findings suggest that plants in symbiosis with rhizobium are more resistant to Cu, the enzyme ACC deaminase may provide an explanation for these results [28]. (Lines 346-347)

Reference:

Kong, Z.Y.; Glick, B.R.; Duan, J.; Ding, S.L.; Tian, J.; McConkey, B.J.; Wei, G.H. Effects of 1-aminocyclopropane-1-carboxylate (ACC) deaminase-overproducing Sinorhizobium meliloti on plant growth and copper tolerance of Medicago lupulina. Plant Soil 2015, 391(1), 383-398.

Ma, W.; Charles, T.C.; Glick. B.R. Expression of an exogenous 1-aminocyclopropane-1-carboxylate deaminase gene in Sinorhizobium meliloti increases its ability to nodulate alfalfa. Appl. Environ. Microbiol. 2004, 70, 5891-589.

Round 2

Reviewer 1 Report

I am glad to review the revised MS from Shen et al., I have carefully reviewed the MS again. It's apparently that the MS has been well-imporved based on my questions. I think the current version can be considered for publication in IJERPH.

Author Response

We very appreciate the reviewer’s positive comments on the current version can be considered for publication in IJERPH. We are glad that the reviewer has agreed with our study.

Reviewer 2 Report

As requested, the authors have provided detailed answers to all the manuscript comments. Unfortunately, the main concern indicated in my previous review, i.e., the improper use of dependent variables in the proposed linear model (variable ~ S + U) as predictors in the relative importance analysis has not been properly addressed. The R relaimpo package allows the analysis of the relative importance of the predictors S (Sinorhizobium inoculation) and U (urea addition) and not the analysis of the relative importance of measured variables that depend on S and U on the observed values of Cu uptake. Cu uptake was chosen as part of a set of variables to measure in response to the experimental treatments. In this experiment, MDA, Soil TN, N, POD, ... cannot be considered environmental drivers of Cu uptake (as I mentioned in my previous comments).  The point is wich factor contributes more to the variation in Cu uptake, MDA, etc.
The authors must present their results according to the experimental design they proposed. This is why I proposed a major revision of the text: in the results section, the authors must indicate if Sinorhizobium inoculation and urea addition had a significant effect on each variable (following the results of the ANOVA analysis). If there is a significant interaction between S and U, then the authors must focus on this interaction effect and not on the main effects. The ANOVA statistics must be reported in the main text (or in tables as in the case of table 2), but not as supplementary material. The relaimpo procedures must be applied to the S and U predictors (contrary to my initial assumtion in my previous comments, the relaimpo package does not manage yet interaction terms, i.e., S:U). Accordingly, the part regarding the relative importance analysis of the measured variables as environmental drivers must be substituted by the relative importance analysis of S and U. The correlation analysis (including the heatmaps) can be maintained because it provides valuable information about the possible mechanisms governing the response of the plants to the experimental treatments.
The discussion section must be rewritten accordingly to the significant changes that must be carried out in the results section. Please focus on the observed interaction effects (or on the main effects if there are not interactions). Replace the comments on the relative importance of MDA, Soil TN, etc. on Cu uptake by comments regarding the correlation among variables (discuss also negative interactions). Finally, in the conclusions section, stress any significant interaction effect (or the absence of interaction effects) between Sinorhizobium inoculation and urea addition, since the novelty of this study relies very much on these hypothesized interactions.       

Author Response

Point-to-point response to reviewer comments

Response to reviewer# 2:

Response: We think that your comments could significantly improve the quality of our manuscript. According to your comments, some related parts have been revised in the revised manuscript. Detailed responses are as below.

Comments 1: Unfortunately, the main concern indicated in my previous review, i.e., the improper use of dependent variables in the proposed linear model (variable ~ S + U) as predictors in the relative importance analysis has not been properly addressed. The R relaimpo package allows the analysis of the relative importance of the predictors S (Sinorhizobium inoculation) and U (urea addition) and not the analysis of the relative importance of measured variables that depend on S and U on the observed values of Cu uptake. Cu uptake was chosen as part of a set of variables to measure in response to the experimental treatments. In this experiment, MDA, Soil TN, N, POD, ... cannot be considered environmental drivers of Cu uptake (as I mentioned in my previous comments). The point is wich factor contributes more to the variation in Cu uptake, MDA, etc.

Response 1: Thank you for your comments and suggestions. We agree with your opinion that the R relaimpo package allows the analysis of the relative importance of the predictors S (Sinorhizobium inoculation) and U (urea addition) and not the analysis of the relative importance of measured variables that depend on S and U on the observed values of Cu uptake. We also agree with you mentioned that MDA, Soil TN, N, POD, ... cannot be considered environmental drivers of Cu uptake. According to your comments and suggestions, we deleted the relative importance (%) of environmental drivers for Cu uptake in plants substituted by analysis of ANOVA and correlation. Our ANOVA results demonstrated that urea and rhizobium had a significant effect on the Cu uptake in both shoots and roots, and there was only an interactive influence on the root Cu uptake. Additionally, our correlation analysis showed that Cu uptake was positively correlated with N content in plants and soil N content (only in shoots). The MDA content was negatively correlated with shoot Cu uptake (Figure 4). The corresponding contents can be seen in the revised manuscript.

Similarly, the Cu uptake in plants was also influenced by urea addition or rhizobium inoculation, but these two factors had no interactive effect on shoot Cu uptake. (Lines 231-232)

The correlation analysis showed that N content was positively correlated with Cu uptake in plants (P < 0.05), and only the shoot Cu uptake was positively correlated with N content in soil (P < 0.01) (Table 7). (Lines 235-237)

Additionally, the MDA content was strongly negatively correlated with Cu uptake in shoots (Figure 4A), probably indicating that rhizobium inoculation through alleviating oxidative stress promoted plant growth thus increasing the Cu uptake. There was no interaction between urea and rhizobium inoculation of OFR (Table 5). And the highest value of OFR in roots was observed with application of urea and rhizobium, indicating that external N supply could induce stress to the inoculated plant roots. (Lines 370-375)

Comments 2: in the results section, the authors must indicate if Sinorhizobium inoculation and urea addition had a significant effect on each variable (following the results of the ANOVA analysis). If there is a significant interaction between S and U, then the authors must focus on this interaction effect and not on the main effects. The ANOVA statistics must be reported in the main text (or in tables as in the case of table 2), but not as supplementary material. The relaimpo procedures must be applied to the S and U predictors (contrary to my initial assumtion in my previous comments, the relaimpo package does not manage yet interaction terms, i.e., S:U). Accordingly, the part regarding the relative importance analysis of the measured variables as environmental drivers must be substituted by the relative importance analysis of S and U. The correlation analysis (including the heatmaps) can be maintained because it provides valuable information about the possible mechanisms governing the response of the plants to the experimental treatments.

Response 2: Thank you for your nice suggestions. According to your comments and suggestions, we further modified the result sections. Firstly, we found that there was a significant influence of urea or rhizobium inoculation on properties of soil and plants (TN, SOM, MDA, SOD, CAT, etc.) as the results of the ANOVA analysis. Secondly, we also found that there was a significant interaction between S and U (based on the ANOVA analysis). Therefore, we focused on this interaction effect by comparing the difference between MSU with other treatments. Finally, according to your suggestion, we maintained the correlation analysis (including the heatmaps) in the revised manuscript. You can find the corresponding ANOVA statistics and correlation analysis in the revised manuscript (Tables 4, 5, and 7; Figure 4). Detailed modifications are in the following responses.

Both urea and inoculation had a significant effect on soil TN content (Table 2), the soil TN content significantly increased by 3.30%, 3.30%, and 7.44% in MU, MS, and MSU relative to M, respectively (P < 0.05). In terms of SOM, urea or inoculation significantly influenced the SOM, and these two factors had an interactive effect (Table 2). (Lines 182-186)

The effects of urea or rhizobium and their interaction on the available Cu in soil were significant, and were enhanced by 4.64% and 15.8% in MS and MSU compared to M, respectively, but decreased by 22.5% after urea addition alone (Table 2). Additionally, only the urea treatment affected the total Cu content. (Lines 187-190)

The addition of urea or rhizobium significantly promoted plant growth and increased biomass both in shoots and roots relative to the control. The urea and rhizobium inoculation had a significant interaction influence on root length, root biomass, and chlorophyll content. (Lines 199-201)

The factors of urea and rhizobium inoculation had an interactive effect on plant N content (Table 4 and 5). In shoots, only inoculation had a significant effect on N content. (Lines 206-208)

Similarly, the Cu uptake in plants was also influenced by urea addition or rhizobium inoculation, but these two factors had no interactive effect on shoot Cu uptake. (Lines 231-232)

In terms of MDA, both urea and inoculation had a significant effect in shoots, and two factors had an interactive effect (Table 4). (Lines 254-255)

The accumulated OFR in differently treated shoots was similar to the control group for neither urea nor rhizobium inoculation had a significant effect (Table 4). (Lines 261-263)

The inoculation had a significant influence on plant POD activity, and urea and rhizobium had an interactive effect (Table 4 and 5). (Lines 273-274)

Both urea and rhizobium had a significant effect on plant CAT activity and these two factors had an interactive effect. (Lines 277-278)

Figure 4 shows the correlations between oxidative damage, Cu content, Cu uptake, TN and antioxidative enzyme activity in plants. In shoots, Cu content showed a strong negative correlation with MDA (P < 0.01) and a positive correlation with POD (P < 0.05). (Lines 294-296)

A negative correlation can be observed between OFR and TN in shoots (P < 0.05). H2O2 was positively correlated with CAT (P < 0.05) and negatively correlated with SOD (P < 0.01). In roots, Cu content showed a strong negative correlation with CAT (P < 0.01). MDA was significantly positively correlated with POD (P < 0.01), and both MDA and POD were strongly negatively correlated with N (P < 0.05). (Lines 298-302)

Comments 3: The discussion section must be rewritten accordingly to the significant changes that must be carried out in the results section. Please focus on the observed interaction effects (or on the main effects if there are not interactions). Replace the comments on the relative importance of MDA, Soil TN, etc. on Cu uptake by comments regarding the correlation among variables (discuss also negative interactions).

Response 3: Thank you for your comments and suggestions. According to your suggestion, the related parts have been rewritten. In the revised manuscript, we focused on the observed interaction effect to emphasize those results. In terms of you mentioned the relative importance of MDA, Soil TN, etc. on Cu uptake has been replaced by correlation among variables. Detailed modifications are in the following responses.

Root biomass in rhizobium-inoculated plants was also higher compared with those treated with urea (Table 3), and there was a negative interaction between urea and rhizobium. This result was in line with their effect on the N content in roots, suggesting that N plays an important role in promoting plant growth (Table 4). Additionally, a negative interaction was observed between urea and rhizobium in chlorophyll content, as evidenced by the fact that a combination of rhizobium inoculation and urea addition decreased plant chlorophyll content more than only inoculation. The ANOVA analysis showed that urea is potentially harmful to inoculated-plant roots. In agreement, previous experimental results showed a decrease in the biomass of soybean nodules when grown with N inferring that N addition reduces rhizobia performance [50]. These findings suggest that plants in symbiosis with rhizobium are more resistant to Cu, the enzyme ACC deaminase may provide an explanation for these results [28]. (Lines 326-336)

In terms of the interaction of urea and rhizobium, there was an inverse influence on Cu content in shoots and roots. Substitutability, the Cu content was stimulated in shoots but decreased in roots with a combination of urea and rhizobium, demonstrated the sensibility of root in response to stress. Lower Cu uptake was observed in roots treated with both urea and rhizobium inoculation relative to inoculation alone (Table 4). There was a negative interaction, is likely due to reduced root colonization from N fertilization [53]. The highest Cu uptake in the roots of urea treated plants (Table 2) can be explained by the fact that NH4+ could change the subcellular distribution (cell wall and vacuole could bind and sequester metals thus limiting their translocation to shoot) [54] and chemical forms of metal (for example, metals integrated with pectates and protein generally have a lower migration) [54, 55]. Additionally, our results showed that plant N was significantly positively correlated with Cu uptake in plants (Figure 4), and soil N was positively correlated with shoot Cu uptake, indicating that the addition of urea or/and rhizobium affected Cu uptake by altering the N content in plant tissues and soil. (Lines 345-356)

Urea application enhanced enzyme activity, including CAT and SOD, suggesting that both enzymes are vital constituents in the defense mechanism of plant shoots (Table 4). There was an interactive effect of urea and rhizobium on Cu content in plant shoots; i.e., Cu content was slightly higher than with inoculation alone. However, POD activity had increased significantly when treated with inoculation alone relative to the combination treatment. This observation suggests that inoculation alone triggers the rise of POD activity in plant shoots. Besides, rhizobium had no effect on shoot SOD activity (Table 4). The above indicated that urea and rhizobium inoculation have a different resistance response to Cu stress in plant shoots. (Lines 385-392)

Meanwhile, our results showed that both urea and rhizobium inoculation had a significant effect on the activities of SOD and CAT in roots (Table 5). Furthermore, compared with the control group, inoculation significantly enhanced the activities of SOD and CAT, indicating that both enzymes play a central role in improving plant root resistance. (Lines 395-398)

Comments 4: Finally, in the conclusions section, stress any significant interaction effect (or the absence of interaction effects) between Sinorhizobium inoculation and urea addition, since the novelty of this study relies very much on these hypothesized interactions.

Response 4: Thank you for your comments and suggestions. We have stressed the significant interaction effect between Sinorhizobium inoculation and urea addition in the conclusion section.

The negative interactive influence was observed between addition of urea and rhizobium, especially in roots, which decreased the N content and increased the oxidative stress more than the rhizobium inoculation alone. Thus we conclude that individual rhizobium inoculation was the most effective method to improve plant resistance in our experiment. (Lines 419-422)
